

# ExhauFS: exhaustive search-based feature selection for classification and survival regression

Stepan Nersisyan[1], Victor Novosad[1,2], Alexei Galatenko[3,4], Andrey Sokolov[3,4], Grigoriy Bokov[3,4], Alexander Konovalov[3,4], Dmitry Alekseev[3,4] and Alexander Tonevitsky[1,2,5]

[1] Faculty of Biology and Biotechnology, HSE University, Moscow, Russia
[2] Shemyakin-Ovchinnikov Institute of Bioorganic Chemistry RAS, Moscow, Russia
[3] Faculty of Mechanics and Mathematics, Lomonosov Moscow State University, Moscow, Russia
[4] Moscow Center for Fundamental and Applied Mathematics, Moscow, Russia
[5] Institute of Nanotechnologies of Microelectronics RAS, Moscow, Russia

Corresponding author
Stepan Nersisyan, snersisyan@hse.ru

## ABSTRACT

Feature selection is one of the main techniques used to prevent overfitting in machine learning applications. The most straightforward approach for feature selection is an exhaustive search: one can go over all possible feature combinations and pick up the model with the highest accuracy. This method together with its optimizations were actively used in biomedical research, however, publicly available implementation is missing. We present ExhauFS—the user-friendly command-line implementation of the exhaustive search approach for classification and survival regression. Aside from tool description, we included three application examples in the manuscript to comprehensively review the implemented functionality. First, we executed ExhauFS on a toy cervical cancer dataset to illustrate basic concepts. Then, multi-cohort microarray breast cancer datasets were used to construct gene signatures for 5-year recurrence classification. The vast majority of signatures constructed by ExhauFS passed 0.65 threshold of sensitivity and specificity on all datasets, including the validation one. Moreover, a number of gene signatures demonstrated reliable performance on independent RNA-seq dataset without any coefficient re-tuning, *i.e.*, turned out to be cross-platform. Finally, Cox survival regression models were used to fit isomiR signatures for overall survival prediction for patients with colorectal cancer. Similarly to the previous example, the major part of models passed the pre-defined concordance index threshold 0.65 on all datasets. In both real-world scenarios (breast and colorectal cancer datasets), ExhauFS was benchmarked against state-of-the-art feature selection models, including $L_1$-regularized sparse models. In case of breast cancer, we were unable to construct reliable cross-platform classifiers using alternative feature selection approaches. In case of colorectal cancer not a single model passed the same 0.65 threshold. Source codes and documentation of ExhauFS are available on GitHub: https://github.com/s-a-nersisyan/ExhauFS.

## INTRODUCTION

Classification algorithms are widely used in biomedical applications, allowing one to construct a rule which will separate data from different classes. For example, classification methods could be applied to determine whether a patient has a particular disease based on gene expression profile (diagnostic test), or to separate patients into groups of high and low risk (prognostic test) (*Cruz & Wishart, 2006*; *Kourou et al., 2015*; *Kang, Liu & Tian, 2016*). In some clinical studies time-to-event data are also available: this means that the data contain not only occurrence or not occurrence of the event, but also the time of observation (*e.g.*, overall, or recurrence-free survival data). Survival regression models, such as Cox proportional hazards model, are used to analyze this type of data (*Zhang, 2002*; *Kleinbaum & Klein, 2012*; *Kamarudin, Cox & Kolamunnage-Dona, 2017*).

One of the main challenges related to machine learning applications in biomedical problems is the so-called "curse of dimensionality": high-dimensional data with a low number of samples often leads to overfitting and poor performance of learning algorithms (*Salsburg, 1993*; *Asyali et al., 2006*; *Sánchez & García, 2018*; *Mirza et al., 2019*). One of the possible approaches to overcome overfitting is called feature selection. Within this framework, some small subset of features is being selected for further classification or regression. Existing feature selection techniques include selection of the most relevant individual features (*e.g.*, the most differentially expressed genes between two classes of interest) and the "native" selection procedures of the most important features from some models, such as random forests, regression techniques with $L_1$-regularization and many others (*Saeys, Inza & Larranaga, 2007*; *Chandrashekar & Sahin, 2014*; *Wang, Wang & Chang, 2016*). Besides that, several approaches were designed specifically for classification problems involving cancer transcriptomics data, including gene ranking, filtration and combining the most relevant genes in a single model (*Arakelyan, Aslanyan & Boyajyan, 2013*; *Zhang et al., 2021*; *Rana et al., 2021*).

Recently, *Galatenko et al. (2015)* proposed to construct classifiers based on all possible feature combinations, *i.e.*, to perform an exhaustive search over feature subsets. Specifically, the authors analyzed all possible gene pairs which allowed them to construct reliable prognostic signatures for breast cancer. Further, the method was successfully applied to the colorectal cancer datasets (*Galatenko et al., 2018b*; *Galatenko et al., 2018a*). Unfortunately, computational complexity of such an approach grows exponentially with the length of tested feature subsets. This means that exhaustive search of all gene triples, quadruples and larger combinations is computationally infeasible even for the most powerful supercomputers. A possible solution for this problem was given in our recent report: the number of features (genes) was preliminarily reduced to allow the search of all possible $k$-element gene subsets, which resulted in the construction of highly reliable eight-gene prognostic signatures for breast cancer (*Nersisyan et al., 2021a*).

In this paper we generalize the latter approach and propose ExhauFS—the user-friendly command-line tool for exhaustive search-based feature selection for classification and survival regression. ExhauFS allows one to vary many algorithm parameters, including classification models, accuracy metrics, feature selection strategies and data pre-processing

methods. Source codes and documentation of ExhauFS are available on GitHub: https://github.com/s-a-nersisyan/ExhauFS.

First, we executed ExhauFS on a "toy" cervical cancer dataset to illustrate the basic concepts underlying the approach. Then, the tool was applied to the problem of 5-year breast cancer recurrence classification based on gene expression profiles in patients' primary tumor. The microarray-based dataset was composed of five independent patient cohorts, the constructed models were also evaluated on the RNA sequencing data from The Cancer Genome Atlas Breast Invasive Carcinoma (TCGA-BRCA) project (*Koboldt et al., 2012*). In order to justify our approach, we tested two commonly used alternative classifier construction pipelines not involving exhaustive search: they were based on univariate feature selection and $L_1$-penalized logistic regression. Finally, we used expression profiles of miRNA isoforms (isomiRs) to build signatures for predicting overall survival in colorectal cancer patients from The Cancer Genome Atlas Colon Adenocarcinoma (TCGA-COAD) project (*Muzny et al., 2012*). As in case of breast cancer, two alternative feature selection methods were benchmarked against ExhauFS: univariate feature selection and sparse Cox model with $L_1$ penalty term.

While near-to-optimal solution can be easily obtained for the first "toy" dataset, the problems solved in the second and the third example are believed to be hard. Specifically, the quality of state-of-the-art solutions in terms of area under the receiver operating characteristic curve (ROC AUC), sensitivity and specificity is well under 1.0 (*Berg et al., 2009*; *Yang, Zhang & Yang, 2019*).

## MATERIALS & METHODS

### Pipeline workflow

The search for predictive models at the top level consists of the following phases. First, a fixed set of features is selected for the downstream processing (feature pre-selection step). Then, $n$ most reliable features are picked according to the feature selection criteria. Next, all possible $k$-element feature subsets are generated from $n$ selected features. For each such a subset, a predictive model (classifier or regressor) is tuned on a training set. Before fitting a model, one can specify data pre-processing transformation. All constructed models are then evaluated on training and filtration sets, user-defined quality metric functions are calculated, and the results are compared with a predefined threshold parameter. In case of successful threshold passage, the model is being saved and evaluated on the validation (test) set.

There is a special part of the tool dedicated to the estimation of running time and, hence, choosing the right values for $n$ and $k$ (so that the pipeline execution takes an appropriate time). The main idea behind the script is to calculate execution time on a limited number of feature subsets and extrapolate this time according to the total number of $k$-element subsets composed from a set of $n$ genes equal to the binomial coefficient $\binom{n}{k}$.

### Feature pre-selection and selection

The first stage of the pipeline is feature pre-selection. The aim of this step is to limit the downstream analysis to a set of user-specified features. The most straightforward and

common way to enable pre-selection is to pass a list of features to ExhauFS through the file. Another useful method is ANOVA $F$-test, which can be used to pre-select features with equal means across training and filtration sets—this option can be beneficial to tackle batch effects (see application example 2 in Results). Besides, pre-selection step can be performed based on some domain knowledge or purpose of a study, *e.g.*, limiting the set of all genes to the list of genes encoding cell adhesion molecules in cancer studies. To validate benefits of using a certain pre-selection method, one should compare reliability of models constructed with and without feature pre-selection.

The next stage is feature selection: features are sorted by some user-specified function, and a certain number of features with the highest scores are picked. Feature selection methods are divided into three main groups: common methods, and methods specific for classification or survival regression problems.

The following feature selection methods are common (*i.e.*, can be used both for classification and regression):

- Taking a ranked list of features from a user-specified file and selecting first $n$ entries.
- Selecting $n$ features with the highest median value across a dataset.

The latter could be potentially useful, for example, in miRNA expression research, where the most expressed entries tend to be the most biologically active ones.

For binary classification problems, the basic feature selection method is Student's $t$-test, whose objective is to score features with the highest differences between means in two considered classes. For multiclass problems we also included ANOVA $F$-test (generalization of $t$-test on three or more groups). Another useful univariate method for data with ordinal class labels consists in calculation of Spearman's correlation coefficients between feature values and class labels, with further selection of features with the highest absolute correlation values. Embedded feature selection methods include $L_1$-regularized logistic regression.

Several available feature selection methods for the survival regression problems are based on the single-factor Cox proportional hazard model. Concordance index and likelihood score can be used to select the most individually informative features. Another group of methods is based on a separation of samples into groups of low and high risk by the median risk score value. In this case, one can use time-dependent AUC, hazard ratio or logrank test $p$-value for feature selection. Similarly to the classification case, one can use sparse $L_1$-penalized Cox models to select features with non-zero weights.

## Data pre-processing and model fitting

Most machine learning methods require data normalization prior to model fitting (*Alexandropoulos, Kotsiantis & Vrahatis, 2019*). A broad set of data pre-processing methods from scikit-learn Python module (*Pedregosa et al., 2011*), including $z$-score transformation, min-max scaler, binarization and others, can be natively used in ExhauFS.

A number of classification models from the scikit-learn module are also natively embedded in ExhauFS interface. This includes support vector machine (SVM), logistic regression, random forest, nearest neighbors and many other classifiers. In addition, one can

use gradient boosting models available in the xgboost package (*Chen & Guestrin, 2016*). It is possible to define both lists of predefined model parameters and parameters which should be estimated through cross-validation (for example, SVM penalty cost parameter or the number of decision trees in a random forest). Survival regression models and metrics were imported from lifelines (https://zenodo.org/record/4816284) and scikit-survival (*Pölsterl, 2020*) modules.

## Implementation

ExhauFS was implemented using Python 3 programming language with an extensive use of common pandas (*McKinney, 2010*), NumPy (*Harris et al., 2020*) and SciPy (*Virtanen et al., 2020*) modules. Open-source code and the detailed documentation can be found on GitHub: https://github.com/s-a-nersisyan/ExhauFS. To make the installation process more convenient, we deposited ExhauFS to Python Package Index (PyPI, https://pypi.org/project/exhaufs/).

ExhauFS also supports parallel execution: each process performs a search on its own set of $k$-element feature combinations. The number of parallel processes used for the pipeline execution is specified by the user. Parallelization is based on the standard python multiprocessing module.

## Datasets used

For the first "toy" example we used the small ($n = 72$) "Cervical cancer" dataset from UCI Machine Learning Repository (*Sobar, Machmud & Wijaya, 2016*). The obtained data were already divided into equal sized training and validation sets.

ER-positive breast cancer microarray dataset (Affymetrix Human Genome U133A Array) was composed of five independent patient cohorts:

- Training set: the union of GSE3494 (*Lundberg et al., 2017*) and GSE6532 (*Loi et al., 2008*) datasets.
- Filtration sets: GSE12093 (*Zhang et al., 2009*) and GSE17705 (*Symmans et al., 2010*).
- Validation set: GSE1456 (*Hall et al., 2006*).

Basic data processing was done like in our previous work (*Nersisyan et al., 2021a*). Briefly, raw *.CEL files were normalized with the use of RMA algorithm available in affy R package (*Gautier et al., 2004*), obtained intensity values were ommllog2-transformed. For each gene we selected a probeset with the maximal median intensity across considered samples. Non-coding and low-expressed genes (lower 25% according to the median intensity values) were discarded. We also removed genes with near-constant expression (difference between 0.95 and 0.05 quantiles lower than two folds). To establish a binary classification problem, we separated all patients into two groups: the ones with recurrence during the first five years after surgery or recurrence-free with at least seven years follow-up. The remaining patients (early censored, lately relapsed or from the grey 5–7 years zone) were included in the construction of Kaplan–Meier plots. For the additional validation, TCGA-BRCA normalized RNA-seq data were downloaded from GDC portal (https://portal.gdc.cancer.gov/) in the format of FPKM expression tables. The ommllog2-transformed FPKM data were converted to the microarray units with the use of quantile

normalization, the useful technique for the gene expression distribution alignment across data samples (*Zhao, Wong & Goh, 2020*). The number of patients in each dataset, as well as the number of patients in each class are available in Table S1.

For the third ExhauFS application example we used isomiR expression data from TCGA-COAD ($n = 413$ patients). MiRNA-seq read count tables were downloaded from GDC portal in the form of *.isoforms.quantification.txt files. Library size normalization was performed with the edgeR v3.30.3 package (*Robinson, McCarthy & Smyth, 2009*), TMM-normalized reads per million mapped reads (TMM-RPM) matrices were generated. The default edgeR noise filtering procedure was applied. We used conventional isomiR nomenclature introduced by *Telonis et al. (2015)*. For example, hsa-miR-192-5p|+1 stands for the mature hsa-miR-192-5p miRNA without the first nucleotide at the 5′-end (the number after the | sign stands for the 5′-end offset in direction from 5′-end to 3′-end). The whole TCGA-COAD cohort was split into three equal-sized sets (training, filtration and validation) with a stratification by outcome and survival time (either date of death or date of the last follow-up). Differential expression analysis for the comparison of tumor and normal samples was performed with the use of DESeq2 R package (*Love, Huber & Anders, 2014*).

For both breast and colorectal cancer datasets, we generated grids of $n$ and $k$ values in such a way that for a specific $n, k$ pair pipeline execution time would not exceed 40 CPU-hours. Feature combination lengths ($k$) were varied between 1 and 20.

All raw and processed data tables, configuration files and instructions for reproducing three presented examples are available on the tool's GitHub page in the "Tutorials" section of the README file.

## RESULTS

### Overview of the approach

We developed an easy-to-use command line tool that combines all necessary parts needed to perform an exhaustive feature selection. The first step of the pipeline is feature pre-selection: the whole downstream analysis is performed for the fixed set of features determined during pre-selection. For example, we recently pre-selected genes encoding extracellular matrix proteins and their cellular receptors and constructed prognostic signatures for colorectal cancer (*Nersisyan et al., 2021b*). The second step of the method is feature selection: features are sorted according to some criteria and then "top $n$" features are selected (*e.g.*, select $n$ most differentially expressed genes between normal and tumor samples).

During the main part, classification or survival regression models are constructed for all $k$-element feature subsets from the set of features selected ($n$, $k$ pairs are selected so that computation time is practically acceptable). Concrete classifiers could be picked from the broad list of available models; optimal user-specified parameters are automatically cross-validated. For survival regression, Cox proportional hazards model is available. Additionally, a variety of data pre-processing methods, such as $z$-score transformation or binarization, could be used.

To tackle batch effects and neutralize effects related to multiple model construction (high-quality classifiers could appear by chance) we introduce filtration sets alongside

training and validation ones. First, model tuning (including cross-validation) is done on a training set for all possible feature combinations. Then, each model is evaluated on one or more special filtration sets: if a pre-defined quality threshold is not met on at least one set, then the model is discarded. Otherwise, if the model demonstrates acceptable performance on both training and filtration sets, it is further evaluated on a validation set. The workflow of the pipeline is presented in Fig. 1, details can be found in Materials & Methods.

The output of the program consists of a table with quality metrics listed for all models which passed the filtration step. To ensure a fair selection of the best model, the rows of the table are sorted by the minimum of accuracy scores for the training and filtration datasets, and the first model is considered as the best one. In case if several models pass the accuracy thresholds, some of them may be chosen instead of the best model based on a relevance to a field of consideration (*e.g.*, combinations of genes with especially important biological roles). Besides, two summary tables are generated. The first one contains the percentage of reliable models (with respect to a validation set) out of all models which passed the filtration step. The second one summarizes the most abundantly occurring features in a set of models which passed the filtration. The percentage of models including a specific feature could be considered as an importance score of the feature. Additionally, one can explore in detail a particular model: this includes construction of receiver operating characteristic (ROC) curves for classification mode and Kaplan–Meier curves for the survival regression mode.

## Application example 1: toy dataset (classification)

As a simple example of how ExhauFS works, we tested it on a small dataset of cervical cancer patients. We searched over all triples of 19 features and found that the best performing random forest classifier was constructed on "perception_vulnerability", "socialSupport_instrumental" and "empowerment_desires" features. This classifier had sensitivity (true positives rate, TPR) equal to 0.9, specificity (true negatives rate, TNR) = 0.96, area under the receiver operating characteristic curve (ROC AUC) = 0.96 (all presented metric were calculated on the validation set). The ROC curve for the classifier is shown in Fig. S1A. In contrast, the performance of the single feature classifiers did not exceed ROC AUC = 0.80 (Table S2).

The output of the program consists of a table with quality metrics listed for all models. To the contrary, using a pure random forest classifier on all available features, we were able to achieve much lower accuracy scores: TPR = 0.8, TNR = 0.92, ROC AUC = 0.92 (Fig. S1B). Interestingly, the set of the most important features in this case ("behavior_personalHygine", "empowerment_knowledge", "perception_severity") did not intersect with the previous one. The model trained only on these three features led to even worse accuracy scores: TPR = 0.6, TNR = 0.88, ROC AUC = 0.83 (Fig. S1C). Thus, the exhaustive search-based approach allowed us to select three features for a random forest classifier which could not be detected by standard "native" feature selection available in the random forest model.

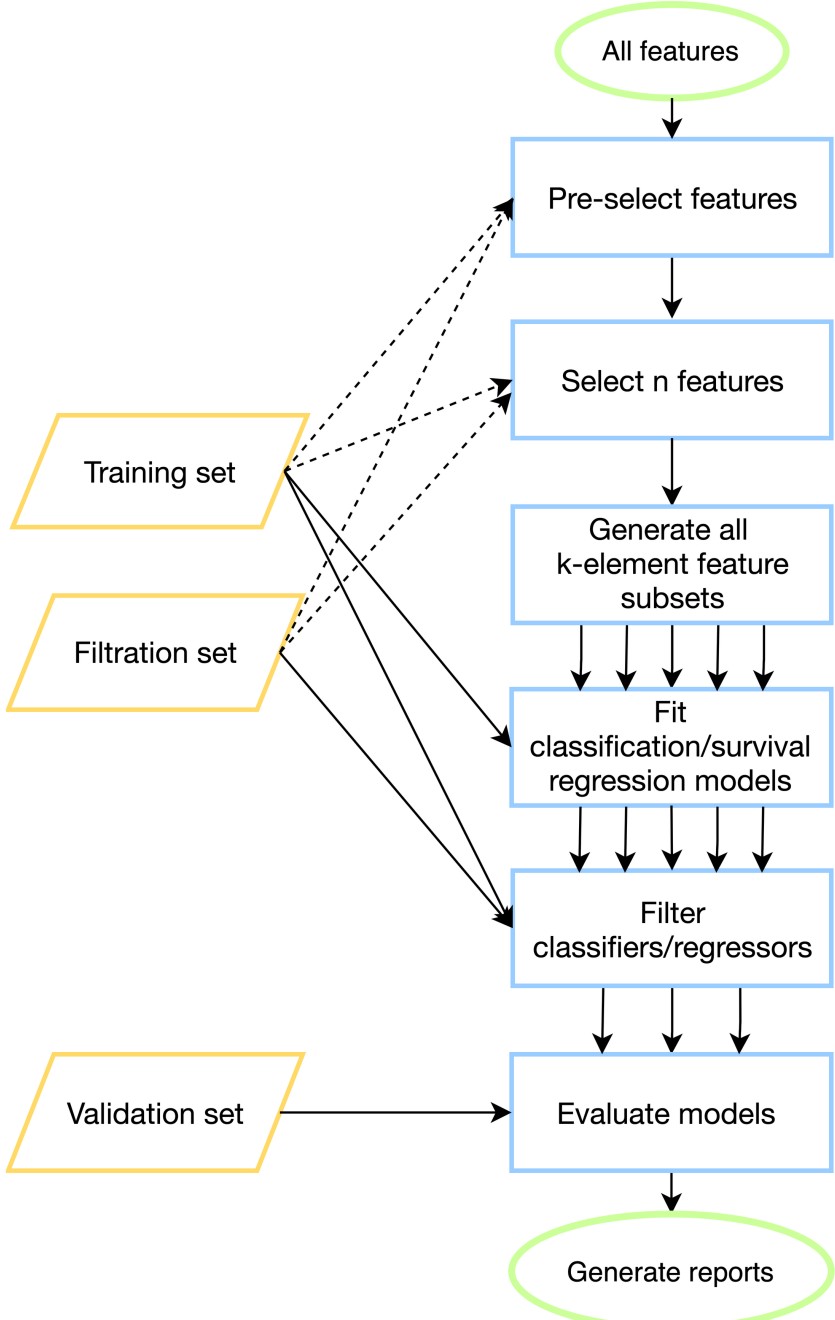

**Figure 1** **ExhauFS workflow.** Dashed lines represent optional relations (*e.g.*, training/filtration datasets can be used in some feature pre-selection methods).

## Application example 2: prognostic gene signatures for breast cancer (classification)

### Experimental setup

We applied ExhauFS to the real-world dataset of gene expression profiles in primary breast tumors (transcriptome profiling was done by hybridization microarrays). The objective was to classify patients into two groups: with recurrence within the first 5 year after surgery or without recurrence with at least 7 years follow-up record. Training, filtration, and validation datasets were composed of five independent patient cohorts (see Materials & Methods).

For feature selection we were picking "top $n$" most differentially expressed genes between classes of patients with and without recurrence. Linear support vector machine (SVM) classifiers were constructed for all $k$ (length of gene signature) in the range from 1 to 20. Prior to model fitting, we applied $z$-score transformation to all genes (*i.e.*, subtracted means and normalized by standard deviations, which were calculated on the training set). Before evaluation on the test set, all trained models were filtered on training and two filtration sets with 0.65 threshold set on TPR, TNR and ROC AUC. Our primary goal was to analyze the percentage of reliable classifiers in a set of models which passed the filtration. For that, we calculated the percentage of classifiers which passed the filtration step and passed the same 0.65 thresholds on the validation set (separately for each $n$, $k$ pair). The closer this value to 100%, the less overfitted the classifier is.

### Execution of ExhauFS on microarray data and pre-selection of "stable" genes

First, we executed ExhauFS without any feature pre-selection. The results of analysis in such a configuration were unsatisfactory. Namely, not a single classifier passed the filtration already for $k \geq 6$ (Table S3). For shorter gene combinations, the maximum percentage of reliable classifiers was only 53% ($k = 3$, $n = 130$). Interestingly, ROC AUC values calculated on the validation set were high for all classifiers which passed the filtration: median ROC AUC $= 0.76$, 95% confidence interval (CI) $0.67 - 0.8$. At the same time, we observed high disbalance between sensitivity and specificity, which could be explained by a presence of batch effects in the data: a decision threshold which is suitable for the training cohort is far from optimal for the validation one. Notably, none of the individual-feature based classifiers ($k = 1$) passed the minimum quality requirements on the training set.

To tackle batch effects and TPR/TNR disbalance, we added the feature pre-selector which was used to put away genes with statistically significant differences in mean values between training and filtration batches (ANOVA $F$-test, validation set was not included). In other words, we preliminarily selected genes with similar expression patterns in training and filtration cohorts ("stable" genes). Inclusion of such a gene pre-selection dramatically increased the quality of the models: thousands of classifiers passed the filtration and more than 95% of them demonstrated high TPR, TNR and ROC AUC values on the validation set for $k \geq 10$ (Table S4). Thus, feature pre-selection is an effective strategy for reducing batch effects between training, filtration and validation sets. Notably, some genes were

highly overrepresented in the constructed gene signatures (example data for $k = 10$ is presented in Table S5).

### Validation of cross-platform classifiers on RNA-seq data

While the major part of existing transcriptomics data of clinical samples with sufficient follow-up periods were generated with microarrays, the current standard transcriptome profiling platform is RNA sequencing. Given that, we assessed whether the classifiers trained on microarray data could demonstrate acceptable performance on RNA-seq data from TCGA-BRCA project.

First, we transformed RNA-seq gene expression values to the microarray units with quantile normalization. Then, the classifiers output by the previous run on microarray data were evaluated on the transformed data without any changes in parameter values (including means and variances of $z$-score scalers and weights/thresholds of SVM models). Eight gene signatures composed of "stable" genes passed the 0.65 sensitivity/specificity threshold. One of the classifiers (a ten-gene signature) had particularly high accuracy: ROC AUC = 0.68, TPR = 0.69, TNR = 0.69 (Fig. 2). Another signature composed of only four genes (*TRIP13*, *ABAT*, *STC2*, *SIGLEC15*) demonstrated slightly worse quality (ROC AUC = 0.66, TPR = 0.67, TNR = 0.69, Fig. S2). Remarkably, sensitivity and specificity scores close to 0.7 are considered to be the current state of the art for the breast cancer recurrence prediction problem (*Van 't Veer et al., 2002*; *Paik et al., 2004*; *Nersisyan et al., 2021a*). To the contrary, not a single classifier passed even 0.6 threshold when no feature pre-selection was applied. Thereby, pre-selection of genes resistant to batch effects allowed us to construct models which could be further generalized to other gene expression profiling platforms.

In all aforementioned cases we used $z$-score transformation for data pre-processing. Though this approach resulted in highly reliable and cross-platform classifiers, the tool allows one to easily vary this step. For example, one can use binarization of gene expression profiles as a pre-processing step: a continuous expression value becomes one if it is greater than a specific threshold (*e.g.*, the median gene expression level) and becomes zero otherwise. The use of such an approach allowed us to achieve a near 100% passability of 0.65 threshold on the validation set when no gene pre-selection was conducted. However, none of the models passed 0.65 accuracy thresholds on the RNA-seq data.

### Comparisons with state-of-the-art feature selection methods justifies reliability of exhaustive search

To compare ExhauFS with alternative approaches, we implemented the canonical classifier construction pipeline: a classifier is fitted directly on selected features (without exhaustive search). Two ubiquitously used feature selection algorithms were tested:

- Univariate feature selection based on the rate of differential expression (Student's $t$-test) (*Samatov et al., 2017*).
- Feature selection based on $L_1$-regularized linear models (logistic regression with $L_1$ penalty term) (*Ma & Huang, 2008*).

In both cases SVM model was fitted on a set of selected features to make an unbiased comparison with ExhauFS. For the same comparison validity reasons, we imposed 0.65

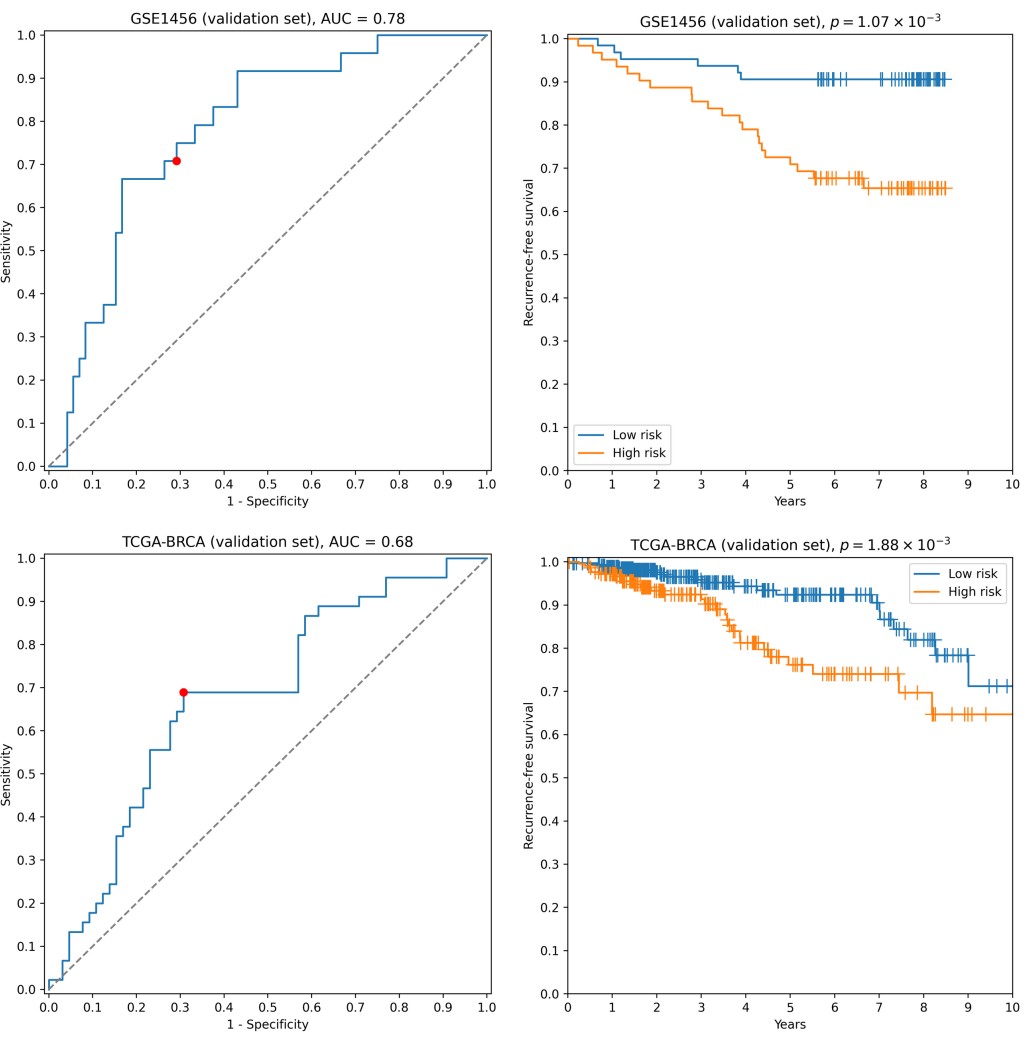

**Figure 2** **ROC and Kaplan–Meier plots for the ten-gene signature (TRIP13, ZWINT, EPN3, ECHDC2, CX3CR1, STARD13, MB, SLC7A5, ABAT, CCNL2) evaluated on GSE1456 (microarray) and TCGA-BRCA (RNA-seq) datasets.** Red points on ROC curves stand for the actual SVM threshold values.

threshold on TPR, TNR and ROC AUC calculated of training and filtration sets. Note that both feature selection methods are available in ExhauFS package, so the proposed pipeline could be reproduced by executing ExhauFS with $n = k$. Besides $k = 1, \ldots, 20$ (which was considered by ExhauFS), we additionally examined values up to 100.

With univariate feature selection only four classifiers with 20, 21, 22 and 23 genes passed 0.65 quality threshold on training and filtration sets. These four models also demonstrated reliable performance on the validation set (minimum of TPR and TNR 0.71). Since this experimental setup was the special case of previously described ExhauFS run (with $n = k$), four identified models were already found by the exhaustive search. Nevertheless, the use of univariate feature selection without subsequent exhaustive search failed to construct shorter cross-platform gene signatures.

With the second feature selection approach (sparse $L_1$-regularized logistic regression), a total of 31 classifiers were selected with 0.65 training and filtration thresholds. Only one model out of them (3.2%) demonstrated acceptable performance on the validation set: 18-gene signature had ROC AUC = 0.79, TPR = 0.67, TNR = 0.76, which was comparable with the best shorter signatures found by ExhauFS. However, quality of predictions for TCGA RNA-seq data was unsatisfactory as opposed to 4- and 10-gene signatures identified by ExhauFS. The detailed information about quality scores for each dataset and gene signature length are presented in Fig. S3.

## Application example 3: prognostic 5'-isomiR signatures for colorectal cancer (survival regression)

Survival regression module of the tool was applied to the problem of predicting overall survival in colorectal cancer patients. For that we used miRNA expression data from TCGA-COAD dataset; miRNA profiling was conducted with small RNA sequencing. Given the fact that even a single nucleotide variation at 5'-end of a miRNA could significantly alter the targetome of the molecule (*Van der Kwast et al., 2019*; *Zhiyanov, Nersisyan & Tonevitsky, 2021*), we consider all possible isoforms of a single miRNA with modified 5'-ends (5'-isomiRs, see Materials & Methods for details about nomenclature).

To analyze the relationship between survival time and miRNA expression levels we used Cox proportional hazard model and concordance index as a measure of model accuracy. Across different feature selectors, the best results were obtained when concordance index was used to select the most individually informative isomiRs. As a result, we identified thousands of isomiR signatures with near-100% passability of 0.65 concordance index threshold on the validation set (Table S6). For example, the seven-isomiR signature (hsa-miR-200a-5p|0, hsa-miR-26b-5p|0, hsa-miR-21-3p|0, hsa-miR-126-3p|0, hsa-let-7e−5p|0, hsa-miR-374a-3p|0, hsa-miR-141-5p|0) had concordance index equal to 0.71, hazard ratio (HR) = 2.97, 3-year ROC AUC = 0.67 and logrank test $p = 8.70 \times 10^{-4}$ (Fig. 3A).

Another successful approach was to use a more biologically motivated feature selection method (*Lv et al., 2020*): to pick the most differentially expressed isomiRs when comparing primary tumors with adjacent normal tissues. While performance of the constructed models was on average lower compared to the previous case (concordance index filtration threshold was dropped to 0.63, Table S7), we were able to identify "short" reliable prognostic signatures. For example, the four-isomiR signature (hsa-miR-126-3p|0, hsa-miR-374a-3p|0, hsa-miR-182-5p|0, hsa-let-7e−5p|0) had concordance index equal to 0.70, HR = 3.43, 3-year ROC AUC = 0.67 and logrank test $p = 6.51 \times 10^{-4}$ (Fig. 3B).

Benchmarking of exhaustive search with alternative methods was conducted in a similar manner as for example 2:

- Cox model with univariate feature selection was directly applied to the data (*i.e.*, $k = n$ special case).
- $L_1$-regularized sparse Cox model was used to select features (*Tibshirani, 1997*), regression model was directly fitted using the selected isomiRs.
- Feature subset length was varied in the range from 1 to 50.

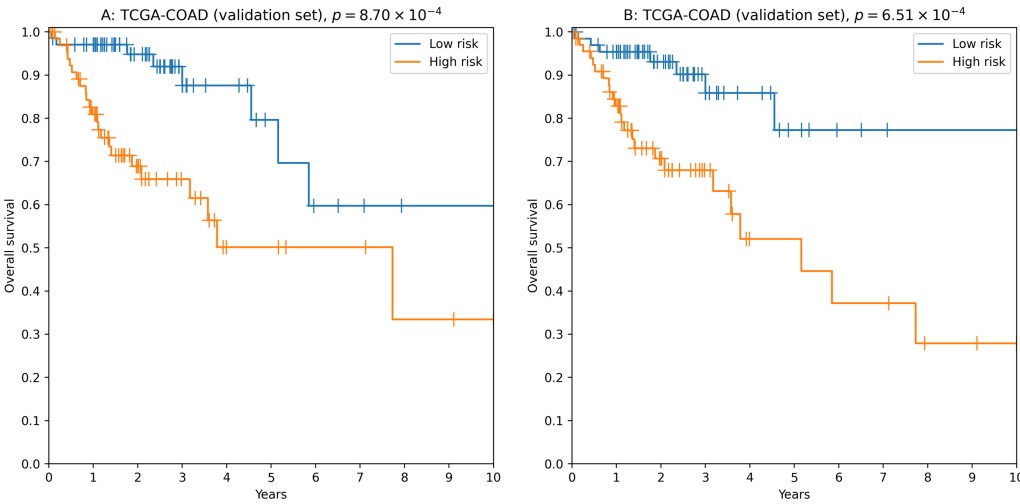

**Figure 3** **Kaplan–Meier plots for 5′-isomiR signatures in colorectal cancer.** (A) The seven-isomiR signature (hsa-miR-200a-5p|0, hsa-miR-26b-5p|0, hsa-miR-21-3p|0, hsa-miR-126-3p|0, hsa-let-7e-5p|0, hsa-miR-374a-3p|0, hsa-miR-141-5p|0). (B) The four-gene isomiR signature (hsa-miR-126-3p|0, hsa-miR-374a-3p|0, hsa-miR-182-5p|0, hsa-let-7e-5p|0).

In both cases, not a single model passed 0.65 concordance index threshold on the filtration set, (Fig. S4). Thus, exhaustive search was a necessary step to construct accurate survival regression models based on 5′-isomiR expression data.

## DISCUSSION

With the use of ExhauFS, one can construct classification and survival regression models for all subsets of selected features. In this manuscript we showed application examples of ExhauFS both on toy and real-world biological datasets and made a comprehensive review on available parameters. The main limitation of our approach consists in high computational complexity of the pipeline: the number of feature subsets grows exponentially relative to the subset cardinality.

The proper use of the algorithm also requires a sufficient number of samples and presence of a validation set. A promising future direction for ExhauFS development includes addition of greedy techniques, which could be used in combination with exhaustive search to expand a set of considered features: see, *e.g.*, the report by *Galatenko et al. (2015)*. Another possible direction is inclusion of network analysis and clustering techniques (*D'haeseleer, 2005*; *Langfelder & Horvath, 2008*) prior to feature selection, which will automatically eliminate clusters of correlated features.

One of the tool application examples was construction of prognostic classification gene signatures for breast cancer. The main challenge of this problem setting consisted in the existence of strong batch effects and biases between independent training, filtration and validation sets. Since tuned models should be further applied to data possibly coming from new batches, conventional batch effect adjustment tools like ComBat (*Johnson, Li & Rabinovic, 2007*) are not applicable in this problem. Given that, we developed a special

feature pre-selection step which allowed us to discard genes which were prone to batch effects. As a result, multiple reliable SVM classifiers were constructed. Moreover, some models showed high accuracy both on microarray and RNA-seq data. Biological reliability of the prognostic signatures was established by the analysis of the corresponding protein functions and previously published cancer reports. For example, three out of four genes from the constructed short cross-platform model (see Results and Fig. S2) were previously mentioned in the context of breast cancer recurrence: *TRIP13* (*Lu et al., 2019*), *ABAT* (*Budczies et al., 2013*; *Jansen et al., 2015*), *STC2* (*Jansen et al., 2015*). While we did not find reports connecting *SIGLEC15* expression with breast cancer recurrence, it was previously shown to play a crucial role in a suppression of anti-tumor T-cell immune responses (*Hiruma, Hirai & Tsuda, 2011*; *Wang et al., 2019*).

Another ExhauFS application example was the construction of survival regression models for colorectal cancer using miRNA isoform (isomiR) expression data. Interestingly, the most reliable isomiR signatures were composed only of canonical miRNA forms. These observations are in agreement with our recent report, where we showed that the majority of 5′-isomiRs are tightly co-expressed to their canonical forms (*Zhiyanov, Nersisyan & Tonevitsky, 2021*). This means that despite functional differences between 5′-isomiRs of the same miRNA, it makes no sense to include both molecules in the machine learning model. Additionally, numerous non-canonical isomiRs were excluded from our analysis because of low expression levels.

## CONCLUSIONS

ExhauFS is a user-friendly command-line tool which allows one to build classification and survival regression models by using exhaustive search of features. A wide set of options can be varied by user, including different algorithms for feature pre-selection, feature selection, data pre-processing, classification and survival regression. Application of ExhauFS to real-world datasets and comparison with alternative pipelines validated the proposed method and its implementation. It is important to note that the scope of ExhauFS applications is not limited to biology-related problems, and it can be used with any high-dimensional data (*e.g.*, text classification or machine learning analysis of financial data).

## ACKNOWLEDGEMENTS

The authors thank Dr. Vladimir Galatenko, Dr. Mikhail Nosov, Yaromir Kobikov and Mokhlaroy Akhmadjonova for valuable comments and discussions.

### Funding

The research was performed within the framework of the Basic Research Program at HSE University. The funders had no role in study design, data collection and analysis, decision to publish, or preparation of the manuscript.

## Grant Disclosures

The following grant information was disclosed by the authors:
Basic Research Program at HSE University.

## Competing Interests

The authors declare there are no competing interests.

## Author Contributions

- Stepan Nersisyan conceived and designed the experiments, performed the experiments, analyzed the data, prepared figures and/or tables, authored or reviewed drafts of the paper, and approved the final draft.
- Victor Novosad performed the experiments, analyzed the data, prepared figures and/or tables, authored or reviewed drafts of the paper, and approved the final draft.
- Alexei Galatenko conceived and designed the experiments, analyzed the data, authored or reviewed drafts of the paper, and approved the final draft.
- Andrey Sokolov, Grigoriy Bokov, Alexander Konovalov and Dmitry Alekseev performed the experiments, authored or reviewed drafts of the paper, and approved the final draft.
- Alexander Tonevitsky conceived and designed the experiments, analyzed the data, authored or reviewed drafts of the paper, and approved the final draft.

## Data Availability

The source codes, files and documentation of ExhauFS are available on GitHub: https://github.com/s-a-nersisyan/ExhauFS.

## Supplemental Information

Supplemental information for this article can be found online at http://dx.doi.org/10.7717/peerj.13200#supplemental-information.

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
