# Peer review of "ExhauFS: exhaustive search-based feature selection for classification and survival regression"

_PeerJ, doi:10.7717/peerj.13200_

## Round 0.1 · original submission · Major Revisions

A minor revision is needed before further processing. Please note that I do not expect you to cite any recommended references unless they are crucial. I look forward to receiving your revision soon.

Reviewer 1 ·

Basic reporting

In this manuscript, Nersisyan et al. reported a novel method to tackle the long-lasting issue of feature selection in practical application, especially the use in dealing with biological data. This research is of high quality, with a clear illustration of the motivation, method, results, and prospective. It is also well-structured with a professional presentation of the figures and results.

Experimental design

This study aims to improve the feature selection method when dealing with large transcriptomic such as gene expression array, and clinical data, such as survival analysis. This research is within the aims and scope of PeerJ. In short, the authors developed a novel method called ExhauFS, which is a feature selection tool based on exhaustive search. I have a few comments on the design:
1, Is any domain knowledge required for the feature pre-selection step? If so, what would be the steps to select the features at the beginning? I imagine that the initial feature set may bias the results, so any randomization (control) method is suggested to show the pre-selection is appropriate?
2, I also suggest the authors pay some attention to the interpretation of the results. For example, when a set of features were selected, how well can they explain the initial hypothesis? If the data contains bias or noise, what additional methods are required to validate the results?
In general, ExhauFS has been tested on multiple datasets and shows promising output compared with current methods. I suggest a minor revision with the above two issues.

Validity of the findings

The authors have tested ExhauFS using multiple datasets by comparing it with current tools. The results also show its robustness in multiple applications focusing on feature selection, including microarray data, survival analysis. The codes of this work have been deposited in GitHub and Python PyPI with a detailed user manual and instructions. The conclusion is also clear and well stated.

Reviewer 2 ·

Basic reporting

This manuscript reported a useful tool, ExhauFS, for classification and survival regression, and authors further used three examples to show the performance of this tool. In my opinion, I think the studies are well-designed and implemented.

Experimental design

Overall good.

Validity of the findings

no comment.

Additional comments

I would like to suggest that the authors discuss the broader potential application of this tool.

Reviewer 3 ·

Basic reporting

Some additional context needed to be provided in the Introduction and the Discussion sections (see Additional comments).

Experimental design

(see Additional comments)

Validity of the findings

no comment in general (see Additional comments)

Additional comments

In the paper, Nersisyan et al. proposed a new feature selection scheme for the identification of gene signatures with a prognostic value. The approach of selection of feature combinations based on an exhaustive search across feature sets of given size seems interesting and demonstrates its power for selecting highly discriminative features.

The authors supplied the paper with all necessary supplemental materials, uploaded their pipeline in the GitHub repository, and provided extensive documentation, which is essential to ensure reproducibility of the paper results as well as its further use.

Overall paper meets the criteria for publication, however, there are several points where clarifications and revisions are needed.

1. The introduction did not to a relevant extent provide a background on the works that were done in this direction. The idea of combining different genes to identify feature sets with higher accuracy has been presented in several research papers and should be discussed in the Introduction section (for example https://ieeexplore.ieee.org/document/6710349, https://doi.org/10.1093/bib/bbaa189, https://www.mdpi.com/2072-6694/13/17/4297, etc.).

2. In the Materials and methods section (line 212-213) it is not clear how what authors mean by transforming FPKM data into microarray units? Did they perform log-transformation of count data or calculated fold-changes?

3. In the “toy” example I suggest authors compare the accuracy of individual features ("perception_vulnerability", "socialSupport_instrumental", and "empowerment_desires") versus constructed classifier containing these three features.

4. In the breast cancer signature authors state that none of the individual genes before preselection did not hit the pre-defines accuracy threshold. However, the authors did not provide the results for the same comparison for genes that passed pre-selection and were used to constrict feature sets. To what extent performance of a single gene is worse compared to their combination?

5. I suggest the authors discuss the possibility of using their approach with only one dataset containing a small number of samples. If dividing the sample into training and validation sets is impossible what other measures and estimates can be used to assess the quality of the selected features.

6. I suggest the authors discuss how to select the one prognostic feature set from many identified ones. For example, in the BRCA example, more than 1000 gene combinations passed the threshold. How to choose one or a few to be used for future comparisons?

7. I also would suggest authors discuss whether signatures identified by ExhauFS perform better or worse compared to the signatures reported previously (not the methods; they did it already in the paper).

8. In Table S1 is not clear what is the total number of patients.

9. Authors report on the SIGLEC15 gene in the discussion, but this gene was mentioned neither in the results section nor in supplementary figures and tables.

---

## Round 0.2 · accepted · Accept

The paper can be accepted. Congratulations!

Reviewer 1 ·

Basic reporting

In this revised manuscript, the authors have successfully addressed my previous concerns. I have read through it and now it meets the publication standard of PeerJ. Thus I suggest acceptance of the manuscript for publication.

Experimental design

The Experimental design meets the standard of the journal.

Validity of the findings

No further improvement is needed.

Reviewer 3 ·

Basic reporting

no comment

Experimental design

no comment

Validity of the findings

no comment

Additional comments

The authors have considered the comment and suggestions. The revised publication can be considered for publication in its current state.